# Investigating the Contribution of Major Drug-Metabolising Enzymes to Possum-Specific Fertility Control

**DOI:** 10.3390/ijms24119424

**Published:** 2023-05-29

**Authors:** Ravneel R. Chand, Mhairi Nimick, Belinda Cridge, Rhonda J. Rosengren

**Affiliations:** 1Department of Pharmacology and Toxicology, School of Biomedical Sciences, University of Otago, Dunedin 9016, New Zealand; mhairi.nimick@otago.ac.nz (M.N.);; 2Science for Communities, Christchurch Science Centre, Christchurch 8041, New Zealand; belinda.cridge@esr.cri.nz

**Keywords:** brushtail possum, *Trichosurus vulpecula*, fertility control, drug-metabolising enzymes (CYP3A and UGT2B)

## Abstract

The potential to improve the effectiveness and efficiency of potential oestrogen-based oral contraceptives (fertility control) for possums was investigated by comparing the inhibitory potential of hepatic CYP3A and UGT2B catalytic activity using a selected compound library (CYP450 inhibitor-based compounds) in possums to that of three other species (mouse, avian, and human). The results showed higher CYP3A protein levels in possum liver microsomes compared to other test species (up to a 4-fold difference). Moreover, possum liver microsomes had significantly higher basal *p*-nitrophenol glucuronidation activity than other test species (up to an 8-fold difference). However, no CYP450 inhibitor-based compounds significantly decreased the catalytic activity of possum CYP3A and UGT2B below the estimated IC_50_ and 2-fold IC_50_ values and were therefore not considered to be potent inhibitors of these enzymes. However, compounds such as isosilybin (65%), ketoconazole (72%), and fluconazole (74%) showed reduced UGT2B glucuronidation activity in possums, mainly at 2-fold IC_50_ values compared to the control (*p* < 0.05). Given the structural features of these compounds, these results could provide opportunities for future compound screening. More importantly, however, this study provided preliminary evidence that the basal activity and protein content of two major drug-metabolising enzymes differ in possums compared to other test species, suggesting that this could be further exploited to reach the ultimate goal: a potential target-specific fertility control for possums in New Zealand.

## 1. Introduction

Brushtail possums (*Trichosurus vulpecula*) are a major concern in New Zealand due to their impact on the country’s biodiversity and economy [1,2]. They are often referred to as ‘reluctant folivores’ because they adapt to different types of vegetation and also consume animal matter such as bird eggs and chicks, insects, and other invertebrates [3]. Possums are also vectors of bovine TB, a disease that affects livestock and has serious consequences for the agricultural industry [4,5]. In 2008, it was predicted that uncontrolled bovine TB would cost the New Zealand livestock industry up to 5 billion NZD over 10 years [2]. For decades there have been continuous efforts (mainly using chemical and physical methods) to reduce the possum population, but the species remains a threat to native biodiversity and the economy, as also recently reviewed by our research group [6].

Synthetic oestrogens in the form of oral contraceptives could serve as a potential tool for possum management in New Zealand. Compounds such as ethinylestradiol (EE2) and diethylstilbestrol can be used alone or in combination with progestin as an oral contraceptive since elevated levels of such synthetic hormones impair the normal functioning of the hypothalamic–pituitary–gonadal (HPG) axis [7]. In particular, EE2 works together with the hormone progestin, which is particularly important for changes in the endometrium of the uterus, to suppress the HPG axis and thus reduce the production of follicle-stimulating hormone (FSH) and luteinising hormone (LH) [8]. The reduced FSH and LH levels thus prevent follicle development and ovulation [9]. This has significantly reduced ovulation in species such as koalas [10,11], mice [12,13], red foxes [14], and white-tailed deer [15].

It is important to note that the use of oral contraceptives for wildlife management must be carefully monitored to ensure their effectiveness and that unintended consequences to animals are avoided. Although oestrogen-based contraceptives have been suggested as a potential tool for wildlife population management, their use can have unintended effects, particularly on behaviour. More specifically, Kidd et al. [16] studied the effects of EE2 on fish populations. Results showed that exposure to EE2 (5–6 ng/L) led to the feminisation of male fathead minnows (*Pimephales promelas*) through the production of vitellogenin mRNA and protein (a female-specific protein). This suggests that exposure to EE2 can disrupt normal sexual development in fish (impair gonadal development and altered oogenesis) and potentially have significant impacts on fish populations. Similarly, Ryan and Vandenbergh [17] observed developmental exposure (sexually dimorphic and non-reproductive behaviour) to EE2 (5 µg/kg/day) in mice. The study found that exposure to EE2 resulted in behavioural masculinisation in all tests used, suggesting that exposure may disrupt normal sexual development in mice by disrupting the neuroendocrine signalling pathways involved in sexual differentiation and behaviour. It should be noted that research on the behavioural effects of oestrogen-based contraceptives on wildlife is very limited, with most studies focusing on aquatic wildlife such as fish [18,19,20]. Likewise, the long-term effects of oestrogen-based contraceptives are not fully understood, and more research is needed to assess their environmental effects along with behavioural effects. However, to support the use of oestrogen-based contraceptives for possums or other pests, more research is needed to understand their potential behavioural and environmental effects, particularly in non-aquatic organisms. More importantly, as an alternative to inhumane and non-selective pest control methods such as using 1080 and traps, oestrogen-based contraceptives could be a useful option.

In addition, although oral contraceptives have successfully prevented ovulation in animals, they also come with the major challenge of rapid hepatic metabolism [21,22,23,24]. The rapid hepatic metabolism of oral contraceptives by drug-metabolising enzymes reduces the amount of synthetic oestrogen and progestin absorbed into the body from the given formulation [24]. The reduced bioavailability of these oral contraceptives can therefore lead to low oral contraceptive efficacy [25]. Previously, as reported for the first time by our group, possum-specific differences in major drug-metabolising enzymes were found compared to rodents (mice and rats). The results showed that possums had ~2-fold higher UGT catalytic activity and ~20% lower CYP3A catalytic activity compared to mice and rats [26]. Therefore, the present study aimed to further investigate possum-specific differences in the major drug-metabolising enzymes (CYP3A and UGT2B) using CYP450-based compounds in rodent, avian and human liver samples. Since oestrogens are metabolised by CYP3A and UGT2B enzymes, finding possum-specific potent inhibitors of these enzymes could help improve the effectiveness of fertility control in possums while limiting their impact on non-target species.

## 2. Results

### 2.1. CYP3A Catalytic Activity

To identify possum-specific activity differences in phase I drug-metabolising enzymes, the basal catalytic activities of CYP3A in mice, possums, avians, and humans were determined (Figure 1A). The basal CYP3A catalytic activity was significantly higher (~2-fold) in mouse than in possum and avian species (*p* < 0.05). While the basal CYP3A catalytic activities between mice and humans and possums and avians were not significantly different (*p* > 0.05). The mean basal CYP3A catalytic activities in these species were 0.57 ± 0.03 (mouse), 0.35 ± 0.01 (possum), 0.34 ± 0.01 (avian), and 0.49 ± 0.04 (human) in nmol/mg/min. In addition, CYP3A4 protein levels also varied between these species (Figure 1B,C). In particular, CYP3A4 protein levels were significantly higher in possums than in avians (~4-fold). While CYP3A4 protein levels were also higher in possums, they were not statistically significant (*p* > 0.05) compared to mice (~2-fold). The mouse CYP3A4 protein levels were ~2-fold higher than the avian. To make a comparison across target and non-target species, mice, possums, avians, and humans were examined together.

Since differences in the basal catalytic activities of CYP3A were found in the test species, a compound library on the structural features of known CYP450 inhibitors was then examined for possum-specific differences in inhibition. These 59 compounds were initially screened with estimated IC_50_ and 2-fold IC_50_ values (Appendix A). Compounds with previously unreported IC_50_ values were screened at concentrations of 1 and 5 μM. From this initial screening, 24 compounds that showed the greatest inhibition against possum CYP3A were selected for further examination. Of these 24 compounds, none showed significant inhibition (<50% of the control) of CYP3A enzyme activity in possum liver microsomes at estimated IC_50_ and 2-fold IC_50_ values or at concentrations of 1 and 5 μM (Appendix A). However, 12 compounds reduced the catalytic activity of possum CYP3A to some extent, mainly at a concentration of 5 μM or estimated 2-fold IC_50_ values, but these reduced catalytic activities were not significantly different from their respective controls (Figure 2A). These compounds were WAY-325945 (65%), ketoconazole (73%), cobicistat (76%), terfenadine (79%), benzbromarone (80%), WAY-657644 (81%), ritonavir (81%), gentiopicroside (82%), WAY-325412 (83%), chlorzoxazone (83%), galeterone (83%), and alizarin (84% of the control). In order to determine species-specific differences, these compounds were then examined in the mouse. Interestingly, ketoconazole was the only compound that significantly reduced mouse CYP3A catalytic activity at IC_50_ and 2-fold IC_50_ values (40–46% of the control, *p* < 0.05) (Figure 2B). Other compounds such as WAY-325945 (67%), cobicistat (68%), 2’-hydroxyacetophenone (75%), terfenadine (83%), ritonavir (84%), and WAY-657644 (88%) showed some level of inhibition in mouse liver microsomes at a concentration of 5 μM or an estimated 2-fold IC_50_ value. However, these compounds were not statistically significant compared to their respective controls. A further six compounds compared to the list of 12 compounds that showed some reduction in possum CYP3A catalytic activities were further examined at a higher concentration (10 μM) in possum and mouse liver microsomes. This was carried out to determine if the tested compounds would further decrease the catalytic activities of CYP3A, particularly in possums. The results showed that these six compounds (alizarin, galeterone, cobicistat, gentiopicroside, chlorzoxazone, and benzbromarone) did not show any significant inhibition of CYP3A catalytic activities in either species at 10 μM (Figure 3).

In addition, 19–21 compounds that showed potential differences in CYP3A inhibitory activity in possums and mice were further examined in avian and human liver microsomes. None of the test compounds showed any significant inhibition (<50% of the control) of catalytic activity in either species. Nonetheless, in avian liver microsomes, most of the compounds showed some level of inhibition at 5 μM concentrations or the estimated 2-fold IC_50_ values, but these did not differ significantly from their respective controls.

### 2.2. P-Nitrophenol Glucuronidation Activity

To identify possum-specific differences in phase II drug-metabolising enzymes, basal UGT2B glucuronidation activity was determined. The possum had significantly higher *p*-nitrophenol glucuronidation activity than the avian (~8-fold) and mouse (~1.5-fold) (Figure 4A). The mouse also had ~5-fold higher *p*-nitrophenol glucuronidation activity than the avian. The basal *p*-nitrophenol glucuronidation activities in these species were 3.60 ± 0.17 (possum), 3.44 ± 0.13 (human), mouse (2.44 ± 0.18), and avian (0.46 ± 0.04) in μmol/mg/min. Additionally, UGT2B4 protein levels also varied between species (Figure 4B,C). The UGT2B4 protein levels from pooled human liver microsomes were higher than those from the avian (~3-fold), possum (~3-fold), and mouse (~2-fold). Furthermore, the mouse hepatic UGT2B4 protein levels were significantly increased compared to the possum and avian.

To examine the possum-specific differences in UGT enzymes, 17 compounds were further screened for their potential to inhibit UGT2B enzymes in possum, mouse, avian, and human liver microsomes (Appendix A). Compounds such as isosilybin (65%), ketoconazole (72%), and fluconazole (74%) showed reduced UGT2B glucuronidation activity in possums, mainly at 2-fold IC_50_ values compared to their respective controls (*p* < 0.05). Even at 2-fold IC_50_ values, these reduced UGT2B inhibitory activities were not less than 50% of the control (Figure 5). When one of these compounds (isosilybin) was further tested for its inhibitory potential on glucuronidation activity with varying concentrations of *p*-nitrophenol (substrate), it also showed no significant changes compared to their respective control activities (Table 1). Similarly, in the mouse, compounds such as fluconazole (78%) and cyclosporin A (80%) reduced *p*-nitrophenol glucuronidation activity at 2-fold IC_50_ values compared to their respective controls (*p* < 0.05), but these inhibitory activities did not fall below 50%. In addition, activity elicited by most of the test compounds in human liver microsomes was over 88% of the control. Even at 2-fold IC_50_ values, there was no major inhibition. Specifically, the compounds screened in human liver microsomes were itraconazole (88%), cobicistat (90%) and fluconazole (94%). In addition, most of the compounds examined showed reduced *p*-nitrophenol glucuronidation activity (ranging from 33 to 87% of the control) in avian species, but these did not differ significantly from their respective controls.

## 3. Discussion

### 3.1. CYP3A Catalytic Activity

The basal catalytic activity of CYP3A was up to 2-fold lower in possum than in mouse species. However, there was no significant difference in catalytic activity between possum–avian and mouse–human. The low catalytic activity of CYP3A in possums may mean that the rate of metabolism of contraceptives in possums is likely to be slower than in rodents and humans. For comparison purposes, it should be noted that there are no previous reports in the literature of basal CYP3A catalytic activity in possums using erythromycin N-demethylation, with the exception of the study previously published by our research team [26]. Our previous study also showed that the basal hepatic CYP3A catalytic activity was approximately 20% lower in possums than in rats and mice. In addition, Hunt et al. [27] reported similar CYP3A catalytic activity (~0.52 nmol/min/mg) in a similar age range (27–60 years) as in the present study in pooled human liver microsomes, while differences (~1.5-fold higher) in the CYP3A catalytic activity have been reported in similar species and age groups of mouse liver microsomes [28]. Furthermore, in the present study, up to 2-fold lower CYP3A catalytic activity was observed in closely related wild mallard species (10 weeks old) compared to avian species [29]. Although differences in CYP3A activity are reported in the literature, there are no studies that have specifically compared possum CYP3A catalytic activity to that of rodents and humans, making it difficult to draw conclusions about those observed in the present. However, it is clear from the present study and our previously published data that possums have a lower rate of CYP3A catalytic activity, and therefore the rate of metabolism of oral contraceptives is likely to be slower than rodents and humans.

Western blotting showed that CYP3A4 protein levels varied between the four species examined. Interestingly, the possum liver had higher levels of hepatic CYP3A4 protein than the avian (~4-fold), mouse (~2-fold), and human (~2-fold), similar to our previous study in possums and rodents [26]. The high CYP3A protein levels in possums could be due to the use of a mammalian CYP3A polyclonal antibody, which may have also detected other highly homologous isoforms of CYP3A in possums. This can be further supported by a previous study that reported that possums express at least three different CYP3A-like isoforms (CYP3A P1, P2, and P3) in their liver and duodenum [30]. However, it is unknown how similarly these CYP3A-like isoforms behave in terms of catalytic activity and subsequently contribute to the observed discrepancy in protein levels and enzyme activity.

A CYP450 inhibitor-based compound library was then screened in test species to identify possum-specific differences in the inhibition of CYP3A catalytic activity. The majority of CYP450 inhibitor-based compounds failed to significantly reduce CYP3A catalytic activity in test species compared to their respective controls. Ketoconazole was the only compound that significantly reduced mouse CYP3A catalytic activity at the estimated IC_50_ and 2-fold IC_50_ values compared to their respective controls. Notably, none significantly reduced possum CYP3A catalytic activity at 1 and 5 µM or the estimated IC_50_ and 2-fold IC_50_ values. However, compounds such as ritonavir, alizarin, galeterone, cobicistat, gentiopicroside, chlorzoxazone, benzbromarone, terfenadine, 7-hydroxyflavone, WAY-325412, WAY-325945, and WAY-657644 showed the potential to reduce the catalytic activity of possum CYP3A to some degree (i.e., reduced activity to ~35% of the control, mainly at 5 µM or an estimated 2-fold IC_50_), but these reduced catalytic activities were not significantly different. None of these compounds (except ketoconazole) have previously been evaluated for their inhibitory effect on CYP3A or UGT2B enzyme activities in possums. Ketoconazole previously inhibited cineole metabolism in wild possum liver microsomes [31]. The same study reported that the formation of all hydroxycineole metabolites (9-, 3-, 7-hydroxycineole) in possums was reduced to approximately 25% of the control by 5 µM ketoconazole. In the present study, ketoconazole (5 µM) similarly reduced the catalytic activity of possum CYP3A, as determined by erythromycin N-demethylation, by approximately 27% compared to the control, but this was not statistically significant, which may be mainly due to inter-individual differences. However, ketoconazole (2.5 and 5 µM) significantly reduced mouse CYP3A by <50% of the control. At similar concentrations, ketoconazole showed only 80% of the control in pooled human liver microsomes. Another screen of ketoconazole at a concentration of 1.25 µM showed a reduced activity of ~64% of the control in pooled human liver microsomes. However, a previous study showed that ketoconazole reduced erythromycin N-demethylation by ~50% of the control at a much lower concentration (0.07 µM) in pooled human liver microsomes [32].

Ritonavir, on the other hand, showed reduced possum CYP3A catalytic activity (81% of the control) at 2-fold IC_50_ (0.28 µM), similar to that in the mouse (84% of the control), but not in humans (95% of the control). Ritonavir has been shown to reduce the metabolism of triazolam (a probe substrate of the CYP3A enzyme) by ~50% at a concentration of 0.014 µM in pooled human liver microsomes [33], but when ritonavir was tested for inhibitory activity at 0.014 µM using erythromycin as substrate, the activity was 95% of the control. In addition, cobicistat also showed a potential reduction in CYP3A activity in all test species: avian (63%), mouse (68%), possum (76%), and human (78%), mainly at 2-fold IC_50_ (or 0.06 µM). Hossain et al. [33] showed that cobicistat reduced the activity of triazolam metabolism by ~50% at 0.032 µM in pooled human liver microsomes. However, the present study found that cobicistat at 0.032 µM reduced erythromycin N-demethylation metabolism to ~78% of the control in pooled human liver microsomes. In addition, Tsujimoto et al. [34] reported that testosterone metabolism in pooled human liver microsomes was inhibited by 7-hydroxyflavone with an IC_50_ > 100 µM. However, our study with 7-hydroxyflavone at estimated IC_50_ and 2-fold IC_50_ values (or 100 and 200 µM) showed activity that was >83% of the control in possums, mice, avians, and humans.

### 3.2. P-Nitrophenol Glucuronidation Activity

This study found that possums had significantly higher basal UGT2B glucuronidation activity than avians (~8-fold). It should be noted that there are no previous reports in the literature of basal UGT2B glucuronidation activity in possums using *p*-nitrophenol glucuronidation, except for the study previously published by our research team [26]. In that study, possums had ~2-fold higher UGT catalytic activity compared to rodents, consistent with the findings of the present study. The higher basal *p*-nitrophenol glucuronidation activity may indicate that possums have a greater ability to metabolise phenolic compounds (including oestrogens) in their liver. This higher *p*-nitrophenol glucuronidation in possum liver microsomes could also explain why marsupial folivores such as the koala can rapidly excrete higher amounts (2–3 g) of glucuronic acids per day than humans (~1000-fold difference) [35,36]. Future studies could therefore focus on examining the role of the possum UGT2B enzymes and their likely association with the greater excretion of glucuronic acids observed in koalas.

In addition, the basal *p*-nitrophenol glucuronidation activity in the mouse was within the range reported in the literature for closely related species: 2.8–2.9 µmol/min/mg in 5–6-week-old female Swiss Webster mouse [37]. However, *p*-nitrophenol glucuronidation activity in humans and avians differed significantly from that reported in the literature. Kuhn et al. [38] reported a lower *p*-nitrophenol glucuronidation activity in pooled human liver microsomes in the range of 0.025–0.38 µmol/min/mg: an approximately 9-fold difference compared to the present study. Similarly, Short et al. [39] reported that *p*-nitrophenol glucuronidation in wild Pekin ducks was 2.55 ± 1.36 nmol/mg/protein/min, approximately 192-fold higher compared to the present study in avian species. No studies in the literature have compared basal UGT2B glucuronidation activity in possums and other test species, but overall, from the present study, the low catalytic activity of CYP3A and the high glucuronidation activity of UGT2B in possums need to be further investigated to understand how these differences in the enzymatic activities of CYP3A and UGT2B enzymes likely contribute to the metabolism and excretion of oestrogen-based contraceptives.

Western blot analysis showed that possum UGT2B4 protein levels were ~3-fold lower than pooled human liver microsomes. The possum UGT2B4 protein levels also differed by ~15% compared to the mouse. Although UGTs are highly homologous across species, their glucuronidation activity and protein content most likely differ due to several factors, as discussed later in Section 3.3. It is therefore important that future research also consider the assessment of such factors in order to understand the extent to which these factors can affect UGT2B activity and protein levels, particularly in possums.

In addition, the compound library used for CYP3A inhibitors was further screened to determine differences in the inhibition of UGT2B activity. None of the compounds screened significantly reduced *p*-nitrophenol glucuronidation activity at estimated IC_50_ values. However, some reduction in possum *p*-nitrophenol glucuronidation activity was mainly observed at much higher concentrations (2-fold IC_50_ values) by compounds such as isosilybin (65% of the control at 200 µM), ketoconazole (72% control at 100 µM), and fluconazole (74% control at 5 mM). None of these compounds have previously been assessed for their potential UGT2B inhibitory effects in possums or other marsupial species. However, these compounds have previously been studied for their UGT inhibitory activities with different enzyme sources and substrate choices in human samples. For example, Gufford et al. [40] reported that silybin B and isosilybin B (both regioisomers of the parent silybin) reduced the glucuronidation of 4-methylumbelliferone (which is known to be metabolised by several UGT isoforms of the UGT1 and UGT2B families) by ~50% of the control in pooled human liver and intestinal microsomes at concentrations of 87 µM and 187 µM, respectively. In the present study, isosilybin showed no significant inhibition of *p*-nitrophenol glucuronidation in pooled human liver microsomes up to 200 µM. In addition, ketoconazole showed a broad UGT inhibitory effect on human Corning supersomes (recombinant forms produced from baculovirus-transfected insect cells) [41]. The results showed that ketoconazole reduced the 4-methylumbelliferone glucuronidation activity of UGT2B4, UGT1A10, UGT1A8, and UGT1A7 by ~50% at concentrations of 12, 20, >100, and 8 µM, respectively. Based on these findings, ketoconazole was screened in the present study for its inhibitory effect on *p*-nitrophenol glucuronidation at 50 and 100 µM. Ketoconazole did not significantly reduce UGT2B activity to less than 50% of the control in any test species at concentrations up to 100 µM. Similarly, other potential inhibitor compounds showed no significant inhibition of *p*-nitrophenol glucuronidation activity at IC_50_ or 2-fold IC_50_ values and thus were not further investigated for their potential possum-specific differences in UGT2B enzymes.

### 3.3. Understanding Discrepancies in Enzyme Activity: Factors Affecting CYP3A and UGT2B Enzymes in Species

The inhibitory activity of screened compounds, as well as the basal catalytic activities and protein content in prepared microsomes of test species on CYP3A and UGT2B enzymes, are important factors in drug metabolism and can have significant implications for drug discovery and development [42,43]. Nonetheless, the discrepancies observed in the inhibitory effect of the compounds examined in this study, in contrast to the literature, can be attributed to various factors.

One significant factor that could account for the observed discrepancies in the major drug-metabolising enzymes observed in the current study versus the literature could be attributed to the choice of substrate. For example, in the present study, ketoconazole at a concentration of 2.5 µM significantly reduced the N-demethylation activity of erythromycin in mice by ~44%. In another study, ketoconazole reduced CYP3A catalytic activity by ~50% at a concentration of 0.0076–0.025 µM with a different set of substrates (midazolam and triazolam) [44]. The difference in substrate choice between the present study and the reported study showed a >100-fold change in the inhibition of the CYP3A enzyme by ketoconazole. Similarly, Stresser et al. [45] reported 2.1- to 195-fold differences in CYP3A inhibitory activity based on substrate choice (benzyloxyresorufin, 7-benzyloxy-4-trifluoromethylcoumarin, 7-benzyloxyquinoline, and dibenzyl fluorescein) in human liver microsomes. These examples therefore suggest that the choice of substrate used to assay the inhibition of drug-metabolising enzymes can significantly affect the results observed. Furthermore, another study suggested that the intricate action of substrates on CYP3A enzymes could be attributed to their interactions within the CYP3A enzyme active site, which could vary depending on the concentration of substrates used and ultimately affect their inhibitory potential [46]. Therefore, it is recommended that future studies use multiple probe substrates at different concentrations to comprehensively assess the inhibitory potential of CYP3A enzymes. Although a single probe substrate specific for CYP3A enzymes was used in the present study to assess species differences, the use of multiple probe substrates would allow for a better understanding of how different substrates affect CYP3A inhibition in in vitro models.

Furthermore, it is worth noting that determining the enzymatic activities of UGTs can present even greater difficulties when compared to CYP3As. This is primarily due to the high homology of UGT enzymes and the overlap of their substrates [47,48]. Even the design and development of fluorescent probes for UGTs is very difficult due to the lack of a full UGT crystal structure, making it difficult to rationally design the candidate substrates [48]. Although UGT2B enzymes were specifically targeted in the present study using *p*-nitrophenol as the probe substrate due to their reported high expression in liver microsomes, it is important to note that *p*-nitrophenol is also known to be metabolised by the UGT1 subfamily [47,49,50,51]. Previous studies have also emphasised the overlapping inhibitory effects of different compounds on UGT2B enzymes. For example, the inhibitory effect of diclofenac, amitriptyline, quinidine, and fluconazole on the enzymatic activities of human UGT2B7 was observed with an IC_50_ between 82 and 2500 μM [52]. However, these compounds also showed inhibitory effects on other UGTs such as UGT1A1, UGT1A9, and UGT1A6. Similarly, bisphenol A (an endocrine-disrupting chemical) showed both competitive (UGT2B4) and non-competitive (UGT2B7, UGT2B15, and UGT2B17) inhibition [53]. Interestingly, this was observed in a recombinant UGT-catalysed 4-methylumbelliferone (4-MU) glucuronidation reaction in vitro with inhibition kinetics (Ki) ranging from 1 to 20 μM. Therefore, caution should be exercised in interpreting the results of UGT enzyme activities, even in the present study, since other subfamilies of UGT enzymes may also contribute to the overall *p*-nitrophenol glucuronidation activity. Future research should aim to develop techniques that can assess the activity of specific UGT isoforms to better understand their role in drug metabolism.

In addition, it is important to note that the use of prepared microsomal samples for different species in the present study may also have resulted in discrepancies in the basal and inhibitory activities of the test compounds for obvious reasons. The discrepancies in the present study could have been influenced by various factors, including but not limited to genetic factors [54,55,56,57,58,59,60,61,62,63], the environment or exposure to xenobiotics [64,65,66,67], and physiological factors [68,69,70,71,72,73,74], all of which can affect the activity of these important drug-metabolising enzymes. It is therefore important to consider such factors and exercise caution when interpreting the results and drawing conclusions about the inhibitory activity of the compounds being screened. This will help to better understand the potential effects of different compounds on drug metabolism, which can influence the drug discovery and development process.

## 4. Materials and Methods

### 4.1. Compound Selection

Bovine serum albumin (BSA), dimethyl sulfoxide (DMSO), nicotinamide adenine dinucleotide phosphate (NADP^+^), and uridine 5′-diphosphoglucuronic acid trisodium salt (UDPGA) were purchased from Sigma-Aldrich (St. Louis, MO, USA). The following primary antibodies were purchased: CYP3A polyclonal antibody (PA5-14896, Invitrogen, Waltham, MA, USA), UGT2B4 (PA5-92155, Invitrogen, Waltham, MA, USA), and GAPDH (Sigma-Aldrich, Sydney, Australia; a subsidiary of Merck KGaA, Darmstadt, Germany, and SHOP, Abcam Australia Pty Ltd., Melbourne, Australia). Fifty-nine CYP450 inhibitor-based compounds (L2000) were obtained from SelleckChem (Houston, TX, USA). These compounds were selected using the SelleckChem database and focused on the key structural moieties of potent CYP3A inhibitors. In particular, the search for compounds focused on heteroatoms/heterocyclic moieties (such as imidazole, pyridine, and triazole) that allow coordination to the iron in CYP3A enzymes [75]. The compound search also included carboxylate ions and aromatic rings. The aim was to improve the interactions between compounds and enzymes via chemical bonds and ionic interactions. Other reagents used were commercially available in the highest purity.

### 4.2. Animals

Possums were collected from the Otago Peninsula (Dunedin) and the Okuti Valley, Banks Peninsula (Christchurch) in New Zealand in collaboration with the Otago Peninsula Biodiversity Group and Manaaki Whenua-Landcare Research, respectively. Possum trapping and liver harvesting were approved by the Peninsula Biodiversity Committee and Manaaki Whenua-Landcare Research. Possums were caged and humanely killed by a blow to the head, and the livers were harvested, snap-frozen in liquid nitrogen, and stored in a freezer at −80 °C until the preparation of microsomal fractions. BALB/c mice (7–9 weeks old) were purchased from Hercus Taieri Resource Unit (Dunedin, New Zealand). Mice were euthanised by carbon dioxide inhalation and the livers were removed and microsomes prepared. The use of mice was approved by the University of Otago animal ethics committee #18/224. The livers from avian species (mallards and a swan) were kindly donated by a local bird hunting group based in Dunedin, New Zealand. The livers from avian species were treated similarly to those mentioned above for the possum samples. Human liver microsomes (HLM) pooled from 50 donors at a protein content of 20 mg/mL were purchased from Thermo Fisher Scientific (Waltham, MA, USA).

### 4.3. Preparation of Mouse, Possum, and Avian Liver Microsomes

Microsomal preparation was carried out as previously described [76]. Individual livers were placed in beakers containing 1.15% KCl. All procedures were performed at 0–4 °C. The liver samples were minced and homogenised in Buffer A (0.1 M Tris, 0.1 M potassium chloride, 1 mM EDTA, and 20 μM butylated hydroxytoluene, at pH = 7.4) using the Teflon-glass homogeniser (5–6 vertical passes). The samples were centrifuged at 10,000× *g* for 20 min. The supernatant was then transferred into another set of centrifuge tubes and spun at 100,000× *g* for 60 min. The pellet was resuspended in Buffer B (0.1 M potassium pyrophosphate, 1 mM EDTA, and 20 μM butylated hydroxytoluene, at pH = 7.4), homogenised, and recentrifuged at 100,000× *g* for 60 min. Finally, the pellet was homogenised in 1–2 mL of Buffer C (10 mM Tris-HCl, 1 mM EDTA, and 4% glycerol (*v*/*v*)), and stored at −80 °C until use. The protein concentration of the microsomes was determined using the bicinchoninic acid (BCA) method [77].

### 4.4. CYP3A Catalytic Activity

CYP3A catalytic activity was determined using the erythromycin N-demethylation activity assay, as described [78]. The reaction mixture (final volume of 1 mL and final protein concentration of 1 mg) contained 500 µL of microsomes in 0.1 M phosphate buffer (pH 7.4) and 400 µL of erythromycin buffer (0.45 mM), and 50 µL of the test compound was pre-warmed at 37 °C for 2 min. The reaction was initiated with 50 µL of 2.5 mM nicotinamide adenine dinucleotide phosphate (NADP^+^) and incubated for 30 min at 37 °C. The reaction was stopped by the addition of 330 µL of 15% zinc sulphate. The tubes were vortexed and left for 5 min (at room temperature), and then 330 µL of saturated barium hydroxide was added. The tubes were further vortexed and left for 5 min (at room temperature). The tubes were then centrifuged at 1800× *g* for 10 min. To 830 µL of the supernatant, 330 µL of Nash reagent (30% (*w*/*v*) ammonium acetate, 0.4% (*v*/*v*) acetylacetone) was added and further incubated for 30 min at 60 °C. The blank tube contained 330 µL of Nash reagent and 830 µL of Milli-Q water. The tubes were recentrifuged at 1800× *g* for 10 min, and the absorbance was read at 415 nm. The CYP3A activities were determined in nmol/mg/min using standard curve data. Results presented are the mean ± SEM of 3 independent experiments performed in duplicate unless otherwise stated.

### 4.5. Glucuronidation of P-Nitrophenol

The rate of *p*-nitrophenol glucuronidation was determined using the optimised protocol [79,80]. Briefly, the incubation mixture (500 µL in total) contained 100 µL of 1 mg/mL microsomes, 50 µL of 1 M Tris-HCl (pH = 7.4), 20 µL of 0.25% (*v*/*v*) Triton-X 100, 50 µL of 50 mM MgCl_2_, 130 µL of Milli-Q water, 50 µL of 5 mM *p*-nitrophenol, 50 µL of the test compound, and 50 µL of 30 mM uridine 5′-diphosphoglucuronic acid trisodium salt (UDPGA). The blank tubes contained all components except UDPGA and microsomes; instead, 150 µL of 0.1 M phosphate buffer was added to equal the volume of the reaction mixture. The reaction mixture and blank were incubated at 37 °C for 2 min. An amount of 50 µL of 30 mM UDPGA was added to initiate the reaction, which proceeded at 37 °C for 30 min. The reaction was stopped using 1 mL of 5% trichloroacetic acid (TCA). The tubes were centrifuged at 1800× *g* for 10 min and then 1 mL of supernatant was added to 250 µL of 2 M sodium hydroxide (NaOH). The absorbance was read at 405 nm. The sample value was subtracted from the blank absorbance and divided by the extinction coefficient of *p*-nitrophenol (18.1 × 10^3^ cm^2^/mole) and the spectrophotometer light path (1 cm). Glucuronidation activities were expressed in µmol/mg/min and are the mean ± SEM of 3 independent experiments performed in duplicate unless otherwise stated.

### 4.6. Western Blotting

The microsomes were diluted to 1 mg/mL in 4X Laemmli buffer (62.5 mM Tris-HCl, 1% sodium dodecyl sulphate, 10% glycerol, 0.005% bromophenol blue, and 355 mM β-mercaptoethanol). The samples were then heated to 95 °C for 5 min and stored at −20 °C. Gel electrophoresis was performed as previously described [81]. All gel casting and running were performed using a Mini PROTEAN^®^ (BioRad). The samples were heated to 37 °C before loading 1–2 µg of protein onto a 7.5% acrylamide gel (acrylamide, 1.5 M lower Tris, 50% glycerol, 10% ammonium persulfate, and tetramethylethylenediamine), and Precision Plus Protein Dual Colour Standards (Bio-Rad Laboratories, Hercules, CA, USA) were loaded as the protein ladder. Empty wells were loaded with 15 μL of 1X sample buffer. The gels were run at 80 V for protein stacking (~15 min) and then at 120 V for protein resolution (~1.5 h). After separation, proteins were transferred to a PVDF membrane (Merck Millipore Ltd., Auckland, New Zealand) in a transfer buffer at 100 V for 60 min. The membranes were blocked in 5% non-fat milk and TBST (Tris-buffered saline with 1% Triton-X 100) for 1 h. The membranes were then incubated with primary antibodies in TBS (Tris-buffered saline) and left on a shaker overnight at 4 °C. The following antibodies were used: CYP3A4 polyclonal antibody (1:1000, PA5-14896, Invitrogen, USA), UGT2B4 (1:1000, PA5-92155, Invitrogen, USA), and GAPDH (1:2000, Sigma-Aldrich, Australia). The membranes were then stripped and washed with TBST buffer (6 × 5 min). Horseradish peroxidase-conjugated secondary antibodies (anti-rabbit for CYP3A4 and UGT2B4 and anti-mouse for GAPDH) in TBS and milk powder were added to the membranes and incubated for 45 min at room temperature. The membranes were re-washed (6 × 5 min) with TBST buffer before the addition of the SuperSignal West Pico chemiluminescent substrate or Femto (ThermoFisher, Albany, New Zealand), and blots were visualised using a CL-XPosure film (ThermoFisher, Albany, New Zealand).

### 4.7. Statistical Analysis

Statistical significance was assessed using a one-way ANOVA with Dunnett’s post hoc test, where *p* < 0.05 was the minimum requirement for a statistically significant difference. All graphing and statistical analyses were performed using GraphPad Prism software version 9.0 for Windows (GraphPad Software, San Diego, CA, USA).

## 5. Conclusions

This is one of the first in vitro studies to investigate how the inhibition of key major drug-metabolising enzymes (CYP3A and UGT2B) in possums could be viewed as a tool to improve the bioavailability of future oral contraceptives for an improved hormonal approach to fertility control. This study confirms that the basal catalytic activity of CYP3A and UGT2B in possums differs from other test species. Specifically, possums had higher CYP3A4 protein levels compared to other test species (up to a 4-fold difference) and significantly higher basal *p*-nitrophenol glucuronidation activity than other test species (up to an 8-fold difference). None of the CYP450 inhibitor-based compounds significantly decreased the catalytic activity of possum CYP3A and UGT2B below the estimated IC_50_ and 2-fold IC_50_ values. Some reduction (>65% of the control) in *p*-nitrophenol glucuronidation activity by compounds such as isosilybin, ketoconazole, and fluconazole has been observed in possums. However, when some of these compounds were tested further, they failed to significantly reduce enzyme activity below the estimated IC_50_ and 2-fold IC_50_ values. There could be many factors that could have influenced the enzymatic activities observed in the present study, some of which may include genetics, enzyme source, preparation, the selection of a probe substrate, or the design of in vitro studies. Furthermore, research could also address the development of techniques to isolate and characterise specific isoforms (or even to develop recombinant enzymes) to selectively study the unique pattern of major drug-metabolising enzymes in possums. Much of the work in this area is still limited and requires further investigation to uncover the details of their contribution to the study of drug metabolism or other enzymes likely to be involved in oestrogen metabolism. Although the literature suggests that these enzymes are highly conserved across species, the development of recombinant forms of these enzymes could aim to provide a thorough understanding for a more effective design of possum-specific inhibitors and for other non-target species.

More importantly, the initial screening of the inhibitory activities of selected compounds may provide insight into the selection of future compounds based on the compound moieties. More specifically, the moieties of three compounds (isosilybin, ketoconazole, and fluconazole) may help narrow future searches for UGT2B4 enzyme-inhibiting compounds that might have similar moieties to achieve possum-specific inhibition of these drug-metabolising enzymes. This finding has the potential to improve oestrogen-based contraceptives, which likely represent a more effective and sustainable method of managing possums, especially when compared to the current widespread use of poisons. The results of this in vitro study have important implications for possum management in New Zealand and beyond. The findings of this study suggest that the management of possum populations could be improved by utilising the insights from this research, potentially providing a more effective and sustainable solution to the ecological and economic challenges posed by possums. Moreover, this study may also present a viable alternative to gene drive technologies, which come with potential unintended consequences and ethical implications. This is particularly important in the context of possum management in Australia, where possums are a protected species, making it unsuitable to adopt a gene drive approach that could risk the extinction of the Australian possum [82]. More importantly, if successful, this study could be adapted for use in other species (such as rodents) to provide a more targeted and effective method of controlling pest populations.

## Figures and Tables

**Figure 1 ijms-24-09424-f001:**
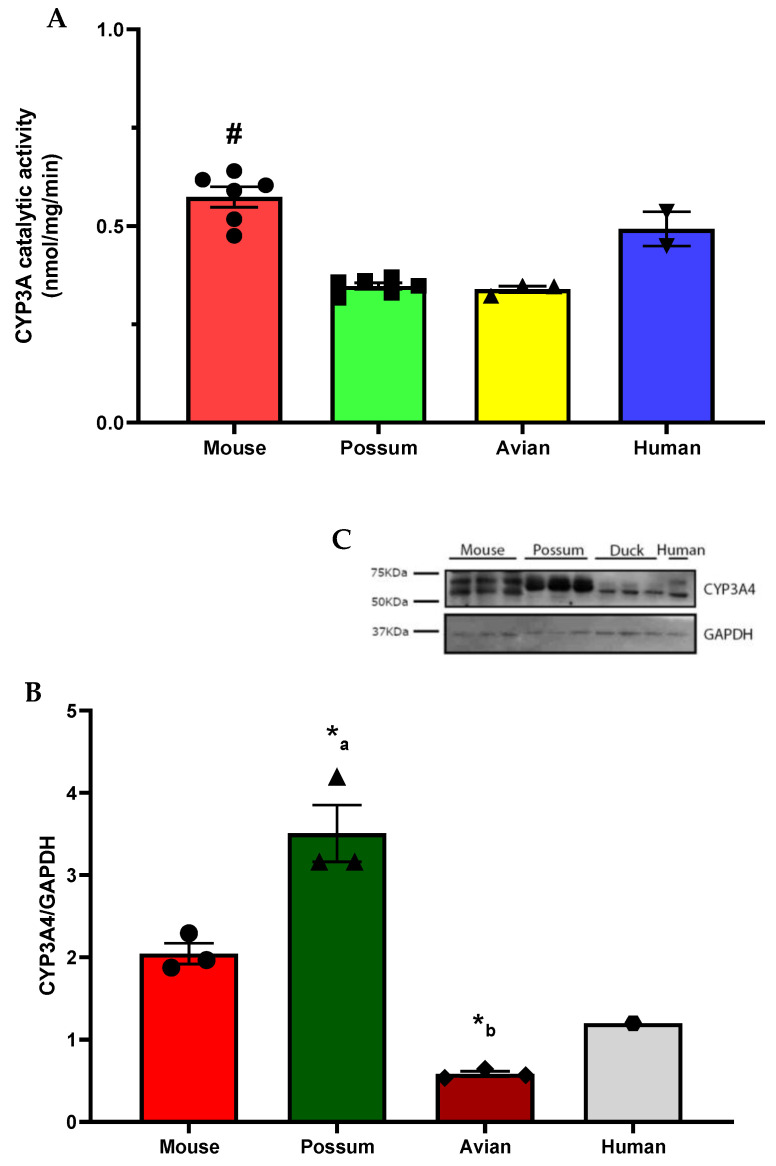
**Basal hepatic CYP3A activity and protein levels in mice, possums, avians, and humans.** Liver microsomes of each species were used to determine CYP3A activity via erythromycin N-demethylation. (**A**). Values are expressed in nmol/mg/min. Bars represent the mean ± SEM of n = 6 in triplicate (mouse and possum), n = 3 in triplicate (avian), and n = 2 in duplicate (pooled human liver microsomes). # Significantly different compared to possum and avian, *p* < 0.05. The protein content for CYP3A4 in liver microsomes from 4 species was determined by Western blotting (load of protein in each well was 2 µg). Hepatic microsomes were also subjected to CYP3A4 Western blotting and the scanning densitometry results were normalised to the housekeeping protein GAPDH. (**B**) The bars represent the mean ± SEM of optical density of n = 3 (except pooled human microsomes, n = 1). (*_a_) Significantly different compared to avians and (*_b_) significantly different compared to mice, *p* < 0.05. Representative Western blot (**C**).

**Figure 2 ijms-24-09424-f002:**
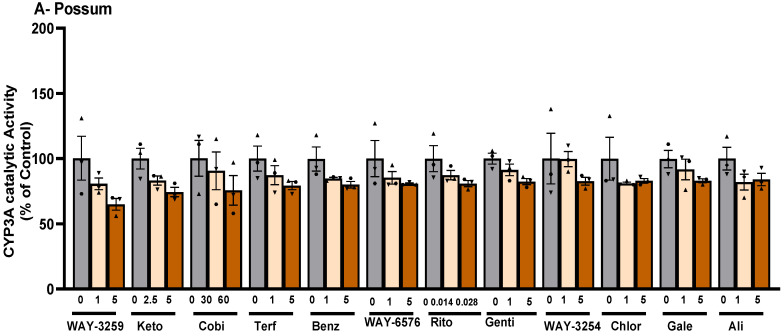
**Effect of CYP450 inhibitor-based compounds on CYP3A catalytic activity in possum and mouse liver microsomes.** Possum (**A**) and mouse (**B**) liver microsomes were incubated with 0.1 M phosphate buffer and CYP450 inhibitor-based compounds, and erythromycin N-demethylation was determined as an indicator of CYP3A inhibitory catalytic activity at estimated IC_50_ and 2-fold IC_50_ values. The dotted line shows the 50% control of CYP3A enzymes. The bars represent the mean ± SEM of n = 3 in duplicate in percent control. * Significantly different from the respective solvent control. Abbreviated compound names refer to WAY-325945 (WAY-3259), ketoconazole (Keto), cobicistat (Cobi), terfenadine (Terf), benzbromarone (Benz), WAY-657644 (WAY-6576), ritonavir (Rito), gentiopicroside (Genti), WAY-325412 (WAY-3254), chlorzoxazone (Chlor), galeterone (Gale), and alizarin (Ali).

**Figure 3 ijms-24-09424-f003:**
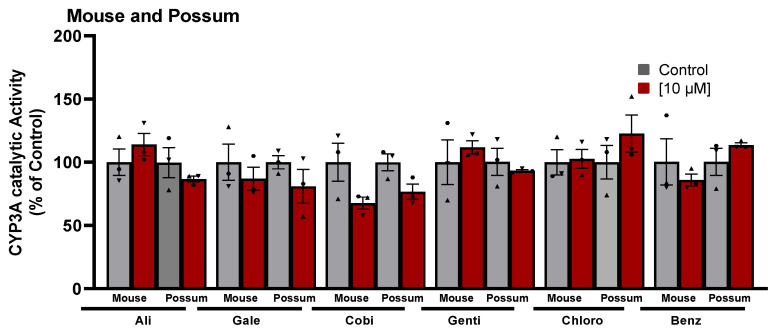
**Effect of CYP450 inhibitor-based compounds on CYP3A catalytic activity in mouse and possum liver microsomes**. Possum and mouse liver microsomes were incubated with 0.1 M phosphate buffer and CYP450 inhibitor-based compounds, and erythromycin N-demethylation was determined at a higher concentration (10 µM) as an indicator of CYP3A inhibitory catalytic activity. The bars represent the mean ± SEM of n = 3 in duplicate. Data were analysed using one-way ANOVA and none were statistically different compared to controls. Abbreviated compound names refer to alizarin (Ali), galeterone (Gale), cobicistat (Cobi), gentiopicroside (Genti), chlorzoxazone (Chlor), and benzbromarone (Benz).

**Figure 4 ijms-24-09424-f004:**
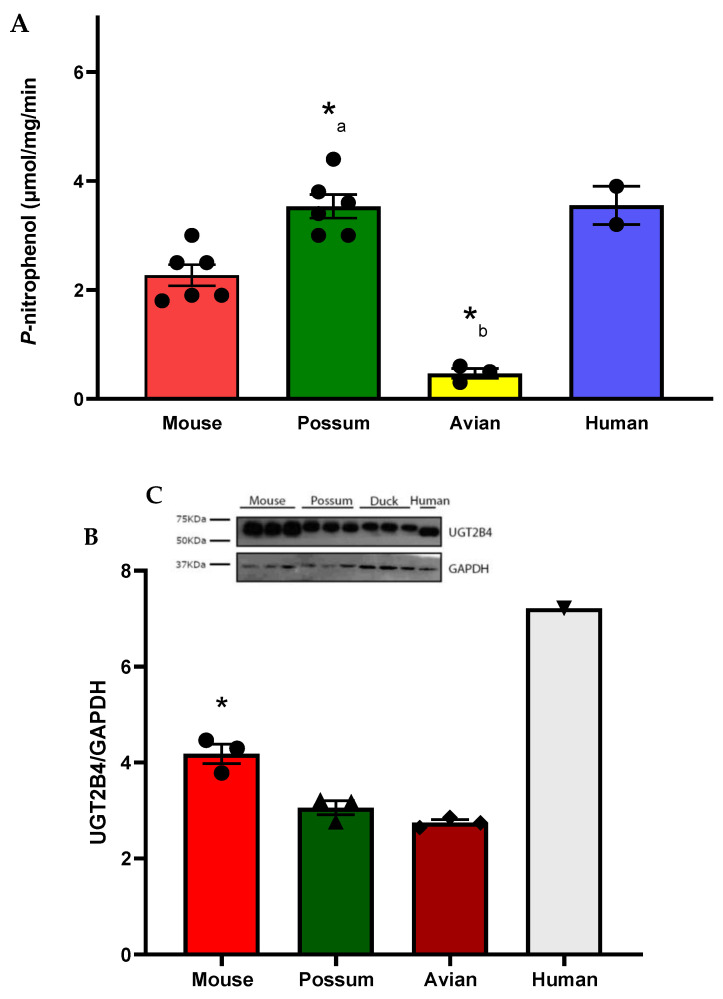
***P*-nitrophenol glucuronidation activity in mouse, possum, avian and human liver microsomes**. Liver microsomes of each species were incubated with 0.1 M phosphate buffer and <10% DMSO, and the *p*-nitrophenol glucuronidation activity was determined as an indicator of UGT glucuronidation activity (**A**). Values are expressed as µmol/mg/min. Bars represent the mean ± SEM of n = 6 in triplicate for possum and mouse, n = 3 in triplicate for avian, and n = 2 in duplicate for pooled human liver microsomes. (*_a_) Significantly different compared to avian and (*_b_) significantly different from mouse, *p* < 0.05. The protein content for UGT2B4 in hepatic microsomes from 4 species was determined by Western blotting (load of protein in each well was 2 µg). UGT2B4 Western blots (**B**) and scanning densitometry results were normalised to the housekeeping protein GAPDH (**C**). The bars represent the mean ± SEM of optical density from n = 3 (except for pooled human liver microsomes, n = 1). * Significantly different compared to solvent control, *p* < 0.05.

**Figure 5 ijms-24-09424-f005:**
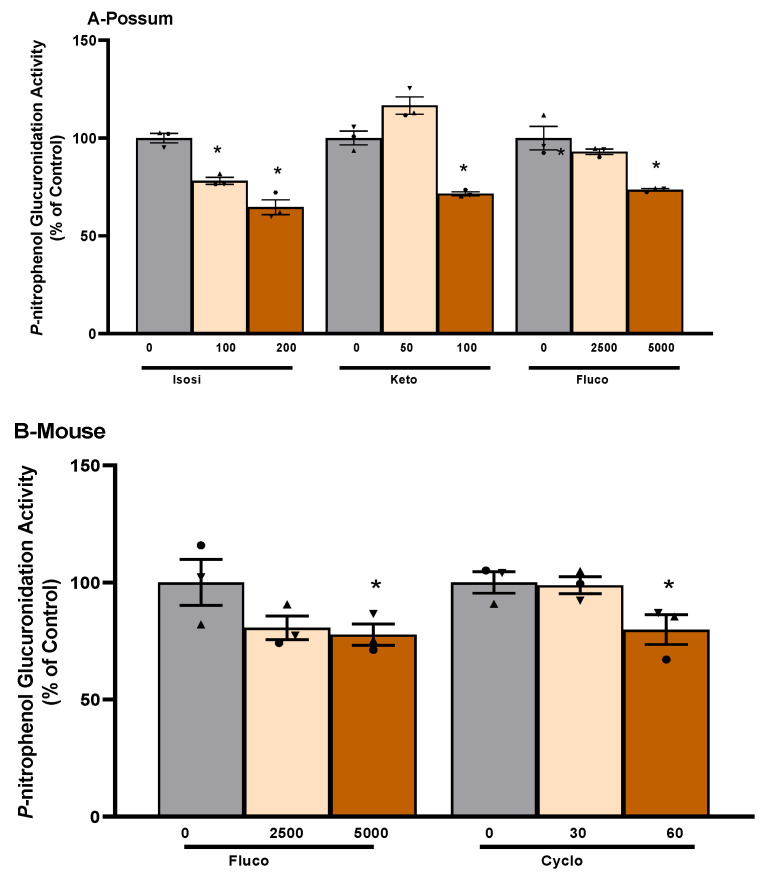
**Compound-mediated inhibition of *p*-nitrophenol glucuronidation activity in possum and mouse liver microsomes.** Liver microsomes of possums (**A**) and mice (**B**) were incubated with 0.1 M phosphate buffer and inhibitory compounds and *p*-nitrophenol glucuronidation activity was determined as an indicator of UGT glucuronidation activity. * Significantly different compared to solvent control, *p* < 0.05. The bars represent the mean ± SEM of n = 3 in duplicate in percent control. Abbreviated compound names refer to isosilybin (Isosi), ketoconazole (Keto), fluconazole (Fluco), and cyclosporin A (Cyclo).

**Table 1 ijms-24-09424-t001:** Substrate-dependent effect of isosilybin on *p*-nitrophenol glucuronidation activity in possum liver microsomes.

Compound	*p*-Nitrophenol Concentration	Concentration (µM)	Glucuronidation Activities ^+^	Percent of Control
**Isosilybin**	2.5 mM	0	0.65 ± 0.15	100
		100	0.47 ± 0.14	75
		200	0.72 ± 0.32	109
**Isosilybin**	1.25 mM	0	0.30 ± 0.07	100
		100	0.15 ± 0.01	56
		200	0.17 ± 0.02	70
**Isosilybin**	0.625 mM	0	0.18 ± 0.02	100
		100	0.12 ± 0.09	63
		200	0.13 ± 0.05	67

^+^ *p*-nitrophenol glucuronidation activity expressed in µmol/mg/min.

## Data Availability

The data presented in this study are reported in the results and Appendix A.

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
