# Peer review of "Investigating the Contribution of Major Drug-Metabolising Enzymes to Possum-Specific Fertility Control"

_ijms, 2023, doi:10.3390/ijms24119424_

Round 1

Reviewer 1 Report

The paper presents an empirical study investigating the contribution of major drug-metabolizing enzymes to  possum-specific fertility control.

The manuscript is well written and understandable.

The research topic may be of relevance for research and practice.

Important strengths include the detailed physiological investigations.

However, some issues could be better clarified.

Since the target concerns fertility control, the ethical aspects of this research need to be discussed in more depth to provide a transparent evaluation.

The risk for negative side effects need to be addressed more attentively.

The practical relevance could be better highlighted. Is this research justified to attain these goals?

Author Response

Thank you sincerely for taking the time to review my paper titled ‘Investigating the contribution of major drug-metabolising enzymes to possum-specific fertility control’. I truly appreciate your valuable feedback and suggestions, as they have greatly contributed to improving the quality and clarity of my work. I have carefully considered each of your comments and made the necessary amendments accordingly. Please find attached the revised version of the paper, which incorporates your insightful input.

Reviewer 2 Report

Overall, the article "Investigating the contribution of major drug-metabolising enzymes to possum-specific fertility control" presents an interesting and well-designed study investigating the potential for estrogen-based oral contraceptives to control the possum population in New Zealand. The study focuses on the contribution of CYP3A and UGT2B catalytic activity to possum-specific fertility control and compares these findings to three other species (mouse, avian, and human).

The study found that possum liver microsomes had significantly higher basal p-nitrophenol glucuronidation activity than the other species tested, and that possums had higher CYP3A protein levels than other test species. However, no CYP450 inhibitor-based compounds significantly decreased the catalytic activity of possum CYP3A and UGT2B below the estimated IC5 and 2-fold ICso values and were therefore not considered potent inhibitors of these enzymes.

The article provides clear and well-presented data, and the conclusions are well-supported by the results obtained. However, there are a few areas for improvement.

First, the introduction could benefit from more background information on possum-specific fertility control and the challenges associated with it. While the introduction briefly mentions the threat posed by possums to biodiversity and the economy, more context on the magnitude of this issue would help readers understand the significance of the study.

Second, the study could benefit from additional data on the potential side effects of using synthetic estrogens as a method of possum-specific fertility control. The study focuses on the potential inhibitory effect of CYP3A and UGT2B catalytic activity, but more information on the risks associated with this method of control would help readers evaluate the feasibility of this approach.

Overall, the study is well-designed, and the results are well-presented and supported. However, the introduction could benefit from more background information, and additional information on the risks of synthetic estrogens as a method of possum-specific fertility control would be helpful.

One minor comment, that the authors only investigated the inhibitory potential of hepatic CYP3A and UGT2B catalytic activity using a selected compound library. Future studies could expand the scope of investigation to include other drug-metabolizing enzymes and a broader range of compounds. Additionally, it would be useful to investigate the impact of these compounds on non-target species to ensure that the fertility control method is specific to possums and does not harm other wildlife.

Author Response

(The authors gave the same response as above.)

Round 2

Reviewer 2 Report

The author has addressed all the concerns raised earlier. Therefore, I kindly request to accept the manuscript for publication.